# Combined Effects of the Thermal-Acoustic Environment on Subjective Evaluations in Urban Park Based on Sensory-Walking

Ye Chen 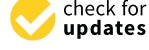, Fan Liu, Xinya Lin, Jing Liu, Ziyi Chen, Kailong Shi, Junyi Li and Jianwen Dong *

College of Landscape Architecture and Art, Fujian Agriculture and Forestry University, 15 Shangxiadian Rd., Fuzhou 350000, China; 2191775001@fafu.edu.cn (Y.C.); 2221775007@fafu.edu.cn (F.L.); 1201775025@fafu.edu.cn (X.L.); 1211775034@fafu.edu.cn (J.L.); 3211726008@fafu.edu.cn (Z.C.); 3211775025@fafu.edu.cn (K.S.); 3191726026@fafu.edu.cn (J.L.)
* Correspondence: fjdjw@fafu.edn.cn

**Abstract:** Studying the impact of various factors on environmental perception is crucial because humans live in an environment where these factors interact and blend. The thermal-acoustic environment is the major factor that affects the overall perception of urban parks. This study focuses on urban parks in the subtropical region, with Xihu Park in Fuzhou, China, as the research area. Through measurements and questionnaires, this study explores the effects of the thermal-acoustic environment in urban parks on subjective evaluation (thermal assessment, acoustic assessment, and overall environmental assessment). The results reveal that: (1) a higher temperature significantly increases the sensation of heat and lowers thermal comfort, heat acceptance, and overall thermal environment evaluation scores. The type of sound source has a significant positive impact on thermal assessment, and the higher the ranking of the sound source type, the greater its positive impact on thermal assessment. (2) Regarding acoustic evaluation, higher sound pressure level is associated with more negative subjective ratings of loudness, harshness, intensity, and excitement. In contrast, positive sound sources can enhance comfort, preference, disorder, coordination, and overall soundscape evaluation. Additionally, temperature increases tend to result in more negative harshness, intensity, and coordination ratings. The interaction between temperature and sound pressure level also significantly affects subjective loudness, harshness, and intensity. (3) Overall environmental evaluation is also affected by temperature, with increasing temperatures leading to decreased comfort and satisfaction while increasing irritation. High sound pressure environments result in worse overall irritation ratings, while positive sound sources can significantly enhance overall comfort, irritation, and satisfaction ratings. Furthermore, the interaction between temperature and sound pressure level significantly impacts overall irritation and satisfaction ratings. These findings are significant for managing and improving the park's thermal environment and soundscape, providing a practical framework for landscape architects.

**Keywords:** urban park; combined effects; temperature; acoustic; subjective evaluations; Fuzhou city; soundscape

## 1. Introduction

With the acceleration of the urbanization process as a phenomenon of the 21st century, a variety of environmental problems have arisen, including water scarcity [1,2], industrial and community waste [3], soil pollution [4], population density increase [5], noise discomfort [6,7], and climate change [8,9]. In this regard, outdoor noise exposure impacts human health, such as hearing loss, sleep disruption, and cardiovascular and mental health disorders [10–12]. On the other hand, the direct impact of climate change on people's quality of life and the emergence of a phenomenon called the Urban Heat Island (UHI) have led to thermal discomfort in urban space [13–15]. Therefore, urban parks are considered an

integral part of the complex urban ecosystem network, providing various advantages for city residents [16]. Open green spaces in urban parks can improve physical and mental health by reducing exposure to air and noise pollution and extreme heat and providing psychological relaxation and stress relief [17].

Despite extensive research on subjective evaluations linking human perception and comfort levels to specific environmental factors in various urban parks, the study of comprehensive influences of different factors remains insufficient [18–20]. As people engage in recreational activities within a multi-faceted, interactive park environment, studying the impact of individual factors on human perception [21] or comfort [22] level fails to represent their genuine environmental evaluations fully. Therefore, it is necessary to investigate the effects of multiple factors on assessing urban park environments.

The soundscape refers to the combination of all sound elements within a specific area, which reflects the cultural, social, and ecological characteristics of that area and can impact individuals' emotional behavior and health status, soundscape should be considered a natural resource that is worthy of management and protection, and therefore suitable measures must be taken [23–26]. Kang (2023) discussed soundscape research's current state and development and explored a framework for using soundscape methodology in urban sound design and planning processes [27]. Francesco, A (2023) integrated smart growth concepts, urban design, and soundscape in a groundbreaking manner. He emphasizes the potential of the soundscape to effectively tackle common noise issues in residential areas [28]. Extensive research has been conducted by scholars from both domestic and international fields on the impact of multiple factors on environmental evaluation [29], particularly on soundscape evaluation [30]. Studies have shown that human visual perception affects auditory perception and the coordination and coherence between sound and image influence preference [31].

Furthermore, sound can serve as supplementary information to visual stimulation [32]. This highlights the crucial role of the relationship between audio and visual elements in creating a comprehensive and effective multimedia experience. With the increase in urban scenery, auditory experiences have become even more complicated. However, research has found that integrating and coordinating auditory and visual elements can enhance aesthetic preferences. Matching soundscapes with visual landscapes can produce a broader and more cohesive sensory experience [33].

Meanwhile, past research has indicated that the thermal environment significantly impacts soundscape evaluation. Factors such as temperature, humidity, lighting, wind, and changes in visual and auditory elements may also affect soundscape assessment, with environmental factors being the most prominent [34,35]. Other scholars have also researched the influence of smell on auditory perception. Through virtual reality experiments, it was discovered that smell could affect the evaluation of road traffic environments, demonstrating its potential to regulate noise and visual landscape perception in specific contexts [36]. Ba and Kang (2019) demonstrated that the presence of smell has almost no impact on the evaluation of bird songs or low-volume sounds, but under certain conditions, a higher concentration of smell leads to more positive evaluations [37]. Several scholars have researched the combined impact of thermal and acoustic environments on physical comfort regarding the comprehensive influence of thermal acoustics. Zhou et al. (2021) discovered that exposure to temperature, noise, and vibration can significantly affect the overall satisfaction of study participants [38].

Regarding the impact of a thermal, acoustic environment on human comfort, Pellerin Nicolas et al. (2003) found that although the combined effect of noise and temperature does not affect physiological data, their research results indicate that noise may alter thermal comfort under warm conditions [39]. Moreover, various factors, such as temperature and noise, have been studied by scholars for their impact on human comfort. Wu et al. discovered that temperature has a significant effect on most subjective and objective parameters, while illumination has a weaker impact. Temperature and illumination are the two primary environmental factors that affect comfort. Furthermore, research has found that

the interaction between color temperature and the temperature has a significant effect on several evaluation parameters [40]. Li et al. (2012) suggest that both temperature and noise satisfaction hold veto power over overall indoor environmental satisfaction, significantly outweighing the importance of lighting satisfaction significantly [41].

A summary of previous research methods indicates that researchers often opt for laboratory or controlled indoor environments as research settings to explore the impact of thermal and acoustic environments on human subjective evaluations. As a result of the relatively stable thermal environment and the ability to effectively control various factors in these venues, currently, there is a lack of research on the subjective evaluation of humans under different thermal-acoustic conditions in real park environments. Urban parks are a significant component of urban construction and an important part of urban ecological systems and city landscapes. They provide a place for urban residents to meet their leisure needs, relax, exercise, socialise, and hold various cultural activities. Therefore, they were chosen as the outdoor research site for this study. Generally, urban parks are located around urban arterials and are often affected by traffic noise at their boundaries.

Additionally, the diverse spatial types within the park and the significant differences in thermal environments influence visitors' environmental evaluations. Therefore, considering factors such as the number of visitors, thermal acoustic environment conditions, and improvement needs, urban parks are conducive to investigating tourists' subjective evaluations under different thermal and acoustic conditions. Finally, it is worth noting that although the activity status of urban park visitors impacts environmental evaluations, in this study, all participants were required to stand or sit [42] when answering the questionnaire. Thus, the instantaneous thermal, acoustic environment has a more significant impact on subjective evaluations.

Considering the limitations of previous research and the necessity of improving the thermal and acoustic environment in urban parks, the present study aims to explore the following aspects: (1) Subjective evaluation of different spaces in the park's environment will be conducted through sensory walks and questionnaires. (2) Objective data such as temperature, global temperature, relative humidity, average wind speed (Va), equivalent continuous sound level (LAeq), as well as dominant sound sources will be collected through monitoring devices to comprehensively understand the park's actual environment and differences in the thermal-acoustic of different areas. This will lay the foundation for the analysis section. (3) Furthermore, this study aims to explore the impact of thermal acoustic on the subjective evaluation of the park. The questionnaires will be analysed to investigate how thermal acoustics affect sound and thermal evaluations as well as overall environmental evaluations of the park. Through these explorations, we hope to provide more scientific suggestions for improving the urban park's thermal acoustics environment.

## 2. Materials and Methods

### 2.1. Study Site Overview

This research was conducted in Fuzhou, a typical city in the humid subtropical region of China. Fuzhou has a pleasant and humid climate with evergreen seasons, abundant sunshine, rainfall, and a short winter and long summer. The frost-free period reaches up to 326 days annually. The average annual sunshine hours are between 1700 and 1980, and the average annual rainfall is between 900 and 2100 mm. The average annual temperature is 20–25 °C, with the coldest month being December and an average temperature of 6–10 °C, and the hottest month being July and August with an average temperature of 33–37 °C. Extreme temperatures range from 2.5 to 42.3 °C annually. The average relative humidity is about 77%, and the city often experiences the urban heat island effect due to its basin topography, where the temperature can reach over 36 °C at noon in summer. The dominant wind direction is northeast, with south wind prevailing in summer [43]. The research site was selected based on whether it had common sound sources and thermal environments in urban parks. To investigate the effect of the thermal, acoustic environment on subjective

evaluations of urban parks, considering factors such as the park area and its accessibility, the Xihu Park, located in the city center, was finally chosen.

Xihu Park is located in the northwest part of Gulou District, Fuzhou City, and is situated at the city's center, adjacent to residential, commercial, and office areas in the city center. It's only 700 m away from the main transportation artery Yangqiao East Road and can be reached by a 10-min walk. With a history of more than 1700 years, it is the most well-preserved classical garden in Fuzhou, and it is renowned as the "Pearl of Fujian Gardens", ranking among the top 36 West Lakes in the country. Currently, it covers an area of 42.51 hectares, with a land area of 12.21 hectares and a water area of 30.3 hectares. The vegetation composition in Xihu Park is complex, dominated by forests, grasslands, shrub beds and flower gardens. The forest area is relatively large, consisting primarily of deciduous and mixed coniferous forests. The deciduous broad-leaved forest in the Huxin Mountain is the most typical, characterized by dominant tree species such as nanmu, camphor, banyan, and paulownia, forming a natural, primitive, and spectacular ecosystem. The grasslands consist of common lawn grass and some wildflowers, while the shrubs comprise flower bushes and climbing plants. The rich and diverse vegetation composition of Xihu Park is functionally complete, providing an ideal place for urban ecological construction and for citizens to relax, exercise and enjoy their leisure time [44].

There are a total of 12 sample sites in Xihu Park. The panoramic view of these sites can be seen in the accompanying Figure 1. In addition, on a representative date of the summer season (5 September), a subjective questionnaire survey will be conducted to monitor the thermal and acoustic environment.

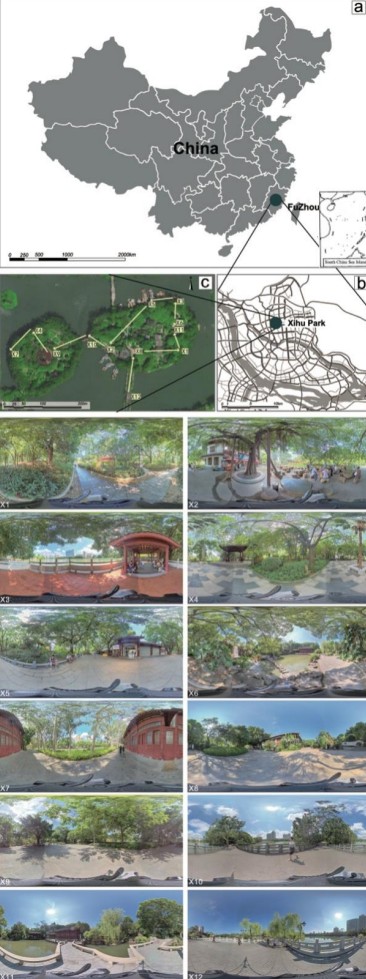

**Figure 1.** (**a**) Location of Fuzhou city in China map; (**b**) Xihu Park in Fuzhou city; (**c**) 12 sample sites within Xihu Park.

*2.2. Procedures*

Regarding the evaluation indicators for the acoustic environment, due to the discontinuity of sound in the park area, this study has opted for the LAeq (weighted equivalent continuous sound level) as the primary evaluation indicator. Moreover, given the variety of sound types in the park, the primary ranking of the dominant sound source type based on the First Impression Sound Source (TP) will be selected as the evaluation indicator through voting. The index ranks in ascending order through positive and negative effects on the sound source, presented as categorical variables [45–49]. As for the evaluation indicators for the thermal environment, due to the instability of the outdoor thermal environment, this study has selected the Universal Thermal Climate Index (UTCI) to evaluate the thermal physiological effects of the thermal environment. The UTCI represents the equivalent environmental temperature of the reference environment and provides the same physiological response as the reference person in the actual environment. This indicator integrates subjective factors and objective environmental parameters while considering the human body's thermal adaptability [50–53]. The UTCI value is derived from the thermal environmental parameters gathered in the questionnaire survey. Once data on radiation temperature, air temperature, relative humidity, wind speed, and global temperature are prepared, the UTCI can be calculated using either the BioKlima2.6 software or the Fortran program available on the official website (http://www.utci.org, accessed on 1 December 2022). The deviation of UTCI from air temperature is determined by actual air temperature ($T_a$), mean radiant temperature ($T_{mrt}$), wind speed ($V_a$), and relative humidity (Rh). This can be expressed by a formula [54].

$$UTCI = T_a + Offset (T_a, T_{mrt}, V_a, Rh) = f (T_a, T_{mrt}, V_a, Rh) \tag{1}$$

In the formula, $T_a$ represents the temperature of a 2 m air column, $V_a$ represents the wind speed of 10 ms, $P_a$ represents the relative humidity, and $T_{mrt}$ represents the average radiative temperature.

*2.3. Questionnaire Survey and Analysis*

Sensory Walk [55] was primarily used as the main research method in this experiment, involving various urban park spatial environments throughout the entire walking route to explore the pattern of acoustic-thermal interaction under different spatial environment conditions. The core of the Sensory Walk is to listen and feel the entire environment. After walking along the designated route and listening to the surroundings, a 5-point Likert scale was adopted to evaluate the thermal, acoustic, and overall environment subjectively. As shown in Tables 1–3, the participants were evaluated based on their subjective loudness vote (SLV), acoustic comfort vote (ACV), acoustic coordination vote (ACoV) [56], acoustic preference vote (APV) [57], acoustic harshness vote (AHV), acoustic pleasure vote (APlV), acoustic familiarity vote (AFV), acoustic intensity vote (AIV), acoustic excitability vote (AEV), acoustic disorder vote (ADV) [58], overall acoustic vote (OAV) [59], thermal sensation vote (TSV), thermal comfort vote (TCV), thermal acceptance vote (TAV), overall thermal vote (OTV) [60], overall comfort vote (OCV), overall annoyance vote (OAnV), and overall satisfaction vote (OSV) [61]. The data were collected to assess various aspects of the acoustic and thermal environment, including subjective evaluation of sound properties, temperature perception, and overall comfort, annoyance, and satisfaction levels. The questionnaire survey as Appendix A.

**Table 1.** Subjective evaluations of the questionnaire surveys (TSV, TCV, TAV, OTV).

| Scores | Thermal Sensation | Thermal Comfort | Thermal Acceptance | Overall Thermal |
|---|---|---|---|---|
| 1 | Cold | Uncomfortable | unacceptable | Bad |
| 2 | cool | Slightly Uncomfortable | Slightly unacceptable | Slightly bad |
| 3 | Neutral | Neutral | Neutral | Neutral |
| 4 | warm | Slightly Comfortable | Slightly acceptable | Slightly good |
| 5 | Hot | Comfortable | acceptable | Good |

**Table 2.** Subjective evaluations of the questionnaire surveys (SLV, ACV, APV, AHV, APlV, AFV, AIV, AEV, ADV, ACoV, OAV).

| Scores | Subjective Loudness | Acoustic Comfort | Acoustic Preference | Acoustic Harshness | Acoustic Pleasure | Acoustic Familiarity | Acoustic Intensity | Acoustic Excitability | Acoustic Disorder | Acoustic Coordination | Overall Acoustic |
|---|---|---|---|---|---|---|---|---|---|---|---|
| 1 | Quiet | Uncomfortable | Unlike | Gentle | Unhappy | Strange | Weak | Calm | Monotonous | Incoordinate | Bad |
| 2 | Slightly quiet | Slightly Uncomfortable | Slightly unlike | Slightly Gentle | Slightly Unhappy | Slightly Strange | Slightly weak | Slightly Calm | Slightly monotonous | Slightly incoordinate | Slightly bad |
| 3 | Neutral | Neutral | Neutral | Neutral | Neutral | Neutral | Neutral | Neutral | Neutral | Neutral | Neutral |
| 4 | Slightly Loud | Slightly Comfortable | Slightly like | Slightly harsh | Slightly happy | Slightly familiar | Slightly strong | Slightly excited | Slightly disordered | Slightly coordinate | Slightly good |
| 5 | Loud | Comfortable | Like | Harsh | Happy | Familiar | Strong | Excited | Disordered | Coordinate | Good |

**Table 3.** Subjective evaluations of the questionnaire surveys (OCV, OAnV, OSV).

| Scores | Overall Comfort | Overall Annoyance | Overall Satisfaction |
|---|---|---|---|
| 1 | Uncomfortable | Peaceful | Bad |
| 2 | Slightly Uncomfortable | Slightly Peaceful | Slightly bad |
| 3 | Neutral | Neutral | Neutral |
| 4 | Slightly Comfortable | Slightly annoying | Slightly good |
| 5 | Comfortable | annoying | Good |

## 2.4. Measurements

The Kestrel 5500 portable weather station was utilized to record temperature, relative humidity, and wind speed. The BES-01 temperature logger was chosen to measure global temperature with a diameter of 0.08m and a surface material scattering coefficient of 0.95. Additionally, the BES-02 temperature and humidity logger was employed to measure air temperature and humidity. The BSWA801 noise and vibration analyzer was utilized to record sound pressure levels in the park environment. All instruments were calibrated before operation. The temperature logger was placed inside a radiation shield to avoid interference from solar radiation and wind.

Meanwhile, the microphone of the analyzer was placed inside a windscreen for more accurate recordings. The instruments were set up according to ISO 772628 [62] and securely mounted on a tripod approximately 1.2 m above ground level. Instrument specifications are detailed in Table 4.

**Table 4.** The characteristics of the instruments.

| Type | Range | Precision |
|---|---|---|
| Kestrel 5500 weather station | 0.4~40 m/s | ±0.1 m/s |
| BES-01 temperature recorder | −30 °C~50 °C | ±0.5 °C |
| BES-02 temperature And humidity recorder | −30 °C~50 °C 0%~99% RH | ±0.5 °C ±3% RH |
| BSWA801 noise vibration analyser | 19 dB(A)~137 dB(A) | <0.7dB (A) |

### 2.5. Subjects

The current study involved 30 subjects, from whom 360 valid questionnaires were collected. All individuals were approached in a park, informed about the purpose of the survey, and voluntarily agreed to participate. Of the subjects, 56.67% were male, and 43.33% were female, with 80% falling between 18 and 30. Moreover, 83.33% (25 individuals) reported residing in Fuzhou for over one year. The subjects had a metabolic equivalent of 1.2 met, and their average clothing insulation was 0.48 clo.

### 2.6. Experimental Process

Firstly, the study's leader provided face-to-face training for all 30 subjects, explaining the basic concept of soundscape, the significance of the experiment, and how to fill out the questionnaire, including the explanation of soundscape questions and answers. Secondly, the leader led the participants through a sensory walk experiment in 12 different areas. The participants were divided into three groups, each supervised by a leader. To avoid overcrowding, there was a 10m distance between each group, and the leading participant controlled the distance [63].

During the sensory-walking experiment, the participants were not allowed to speak, eat or drink to reduce human noise. Finally, they walked through each designated sensory walking area along a specified route. The group stopped at each site for five minutes, standing or sitting while carefully listening to sound elements and feeling the thermal environment, then filled out a questionnaire online. The experimental process is shown in Figure 2. The primary leader was also responsible for using equipment to collect data, including temperature, global temperature, humidity, wind speed, and the LAeq (a weighted equivalent continuous sound level), for a continuous period of 5 min [64].

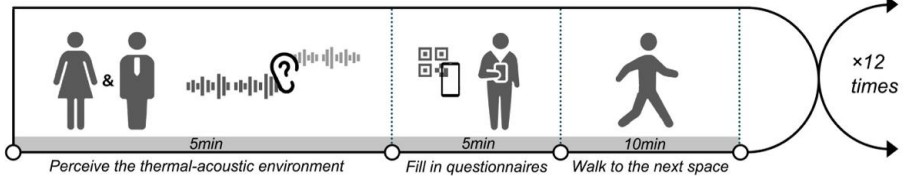

**Figure 2.** Experimental process.

### 2.7. Analysis Method

The experimental results were free of missing values, and any outliers were eliminated or replaced to ensure the accuracy and reliability of the data. Following data collection and organization, objective parameters were standardized to calculate each variable's mean and standard deviation.

In both production activities and scientific experiments, the generation of results is often influenced by multiple factors with varying degrees of impact. ANOVA, a statistical method, infers whether one or more factors have a significant impact on the experimental results when they change at different levels. The study utilized the SPSS27.0 statistical analysis software to analyze subjective perception data of the environment. The variance analysis method (ANOVA) was employed to explore the differences in the impact of thermal-acoustic elements on subjective perception, and the Mauchly sphericity test was conducted. When the results did not satisfy the sphericity assumption ($p < 0.05$), a correction was needed. The Greenhouse-Geisser correction was chosen when Epsilon < 0.75, while the Huynh-Feldt correction was selected in other situations [65]. The final experimental data passed validation before analysis.

## 3. Results

### 3.1. Basic Environmental Conditions

The researchers conducted a sensory walk experiment in 12 different areas under the guidance of local experts and research teams, following a pre-established plan. Table 5 illustrates the basic thermal-acoustic data acquired through equipment in the 12 areas.

**Table 5.** Sample thermal-acoustic environment situation.

| Plot | Ta (°C) | Ts (°C) | Humidity (%) | Va (m/s) | UTCI (°C) | LAeq (dB) | Source Type |
|------|---------|---------|--------------|----------|-----------|-----------|-------------|
| 1 | 30.30 | 30.50 | 57.50 | 0.00 | 31.35 | 53.90 | 2 (bird song) |
| 2 | 31.10 | 31.20 | 54.40 | 0.40 | 32.16 | 64.60 | 5 (conversations) |
| 3 | 31.40 | 31.70 | 55.60 | 0.00 | 32.50 | 58.60 | 5 (conversations) |
| 4 | 31.60 | 31.80 | 70.10 | 0.20 | 34.68 | 52.90 | 3 (rustling leaves) |
| 5 | 32.00 | 33.10 | 68.20 | 0.00 | 35.03 | 61.10 | 4 (broadcasting music) |
| 6 | 31.90 | 32.80 | 73.10 | 0.10 | 35.77 | 71.50 | 1 (water) |
| 7 | 32.30 | 33.50 | 68.20 | 0.30 | 35.89 | 51.10 | 2 (bird song) |
| 8 | 32.60 | 33.50 | 68.10 | 0.30 | 36.16 | 59.40 | 4 (radio music) |
| 9 | 32.80 | 34.60 | 65.20 | 0.10 | 36.19 | 54.20 | 2 (bird song) |
| 10 | 33.80 | 35.50 | 62.60 | 0.80 | 37.56 | 59.50 | 4 (broadcasting music) |
| 11 | 34.10 | 36.20 | 65.70 | 0.10 | 38.26 | 63.50 | 1 (water) |
| 12 | 34.80 | 36.80 | 61.00 | 0.60 | 38.92 | 77.80 | 6 (Construction noise) |

### 3.2. Comprehensive Impact of Thermal Acoustic Environment on Thermal Assessment

The impact of the thermal and acoustic environment on thermal assessment includes the effects on TSV, TCV, TAV, and OTV. Table 6 demonstrates the significance of the indicators under main effects and interaction effects (TSV, TCV, TAV, and OTV). The analysis of variance (ANOVA) indicates that the main effects of UTCI significantly influence TSV ($p < 0.01$), TCV ($p < 0.05$), TAV ($p < 0.05$), and OTV ($p < 0.1$). On the other hand, the interaction effect between LAeqUTCI and LAeq has no significant impact on all thermal assessments ($p > 0.1$), while the main effects of TP significantly influence TSV, TCV, and OTV ($p < 0.01$).

**Table 6.** The significance of the indicators under the main effect and interaction (TSV, TCV, TAV, and OTV).

| Subjective Evaluation | UTCI | LAeq | UTCI × LAeq | TP |
|-----------------------|------|------|-------------|-----|
| TSV | **0.003** | 0.196 | 0.12 | **0** |
| TCV | **0.016** | 0.298 | 0.221 | **0.001** |
| TAV | **0.039** | 0.219 | 0.155 | 0.225 |
| OTV | **0.059** | 0.611 | 0.458 | **0.007** |

Note: Bold indicates significant correlation within 0.1.

Figure 3 illustrates the TSV values of 12 sample sites. Concerning the impact of the thermal environment on TSV, there is a significant increasing trend in TSV as UTCI increases. When UTCI is lower than 36, TSV is concentrated in the moderate range, while the deviation of TSV is greater when UTCI is above 36. As for the influence of noise environment on TSV, LAeq has little effect on TSV as a whole. However, when UTCI is below 34 or above 37, TP can affect TSV and the degree of deviation increases.

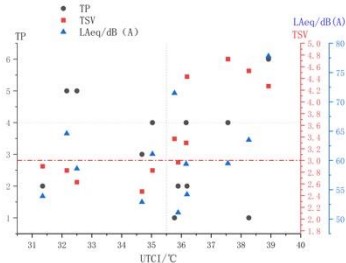

**Figure 3.** The values of TSV under different conditions of the thermal-acoustic environment.

Figure 4 demonstrates the TCV values of the 12 sample sites. Regarding the impact of the thermal environment on TCV, there is an obvious decreasing trend in TCV as UTCI increases. When UTCI is around or below 35.5, TCV is relatively comfortable, and discomfort rises when UTCI exceeds 36. As for the impact of noise environment on TCV, LAeq has little effect on TCV as a whole. Nevertheless, positive TP can increase TCV when UTCI is below 36.

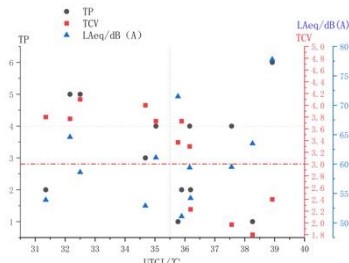

**Figure 4.** The values of TCV under different conditions of the thermal-acoustic environment.

Figure 5 illustrates the TAV values of 12 sample sites. Regarding the impact of the thermal environment on TAV, there is a significant decrease in TAV with the increase of UTCI. When the UTCI value is below 36, the TAV score is higher, while the degree of thermal comfort significantly reduces when the UTCI value exceeds 36. As for the impact of the acoustic environment on TAV, LAeq and TP have no significant influence on TAV as a whole.

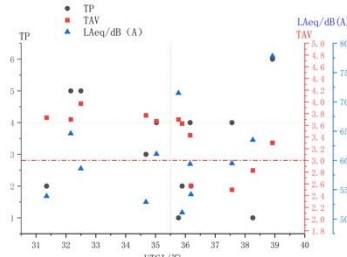

**Figure 5.** The values of TAV under different conditions of the thermal-acoustic environment.

Figure 6 displays the OTV values of 12 sample sites. Concerning the impact of the thermal environment on OTV, there is a significant decrease in OTV with the increase of UTCI. When the UTCI value is below 36, the OTV score is higher than neutral, but it falls below neutral when the UTCI value exceeds 36. As for the impact of the acoustic environment on OTV, LAeq has no significant influence on OTV as a whole. However, when UTCI is around 35.5, the positive TP can increase the OTV score.

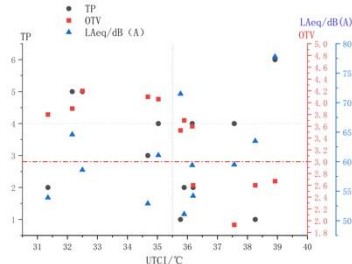

**Figure 6.** The values of OTV under different conditions of the thermal-acoustic environment.

### 3.3. Comprehensive Influence of Thermal Acoustic Environment on Acoustic Assessment

To ensure the consistency of the results and avoid misleading situations, the inversion of the rating procedure is adopted before statistical analysis, with the parameters of ACV, APV, AFV, and OAV being flipped [66]. The impact of the thermal, acoustic environment on acoustic assessment includes its effect on 11 indicators, namely, SLV, ACV, APVA, HVA, PlVAF, VAIV, AEV, ADV, ACoV and OAV. Table 7 demonstrates the significance of the main effects and interactions of these indicators. Variance analysis reveals that UTCI has a significant main effect on AHV ($p < 0.1$), AIV ($p < 0.01$) and AEV ($p < 0.1$). LAeq has a significant main effect on SLV ($p < 0.01$), AHV ($p < 0.01$), AIV ($p < 0.01$) and AEV ($p < 0.05$). The interaction effect of UTCI and LAeq has a significant impact on SLV ($p < 0.05$), AHV

($p < 0.05$), AIV ($p < 0.01$), and AEV ($p < 0.05$). Meanwhile, TP has a significant main effect on ACV, APVA, DVO and OAV ($p < 0.1$).

**Table 7.** The significance of the indicators under the main effect and interaction (SLV, ACV, APVA, HVA, PlVAF, VAIV, AEV, ADV, ACoV and OAV).

| Subjective Evaluation | UTCI | LAeq | UTCI × LAeq | TP |
|---|---|---|---|---|
| SLV | 0.133 | **0.007** | **0.034** | 0.12 |
| ACV | 0.209 | 0.129 | 0.133 | **0.001** |
| APV | 0.699 | 0.474 | 0.513 | **0.019** |
| AHV | **0.086** | **0.007** | **0.027** | 0.418 |
| APlV | 0.577 | 0.394 | 0.441 | 0.26 |
| AFV | 0.969 | 0.858 | 0.887 | 0.581 |
| AIV | **0.006** | **0** | **0.001** | 0.222 |
| AEV | **0.058** | **0.014** | **0.034** | 0.137 |
| ADV | 0.518 | 0.417 | 0.333 | **0.002** |
| ACoV | **0.093** | 0.135 | 0.147 | **0.042** |
| OAV | 0.724 | 0.748 | 0.739 | **0.082** |

Note: Bold indicates significant correlation within 0.1.

Figure 7 illustrates the SLV values of 12 different sample sites, showing a significant increase in SLV with an increasing LAeq due to the influence of noise. When the LAeq exceeds 53 dB(A), SLV values are consistently higher than the neutral level. In terms of the impact of the thermal environment on SLV, the UTCI has no significant effect on the overall SLV.

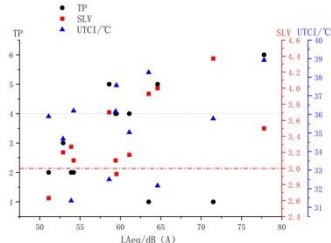

**Figure 7.** The values of SLV under different conditions of the thermal-acoustic environment.

Figure 8 displays the ACV values of the same 12 sample sites, indicating that the ACV value decreases to some extent when the TP is below four due to the influence of noise. Regarding the impact of the thermal environment on ACV, the UTCI has no significant effect on the overall ACV.

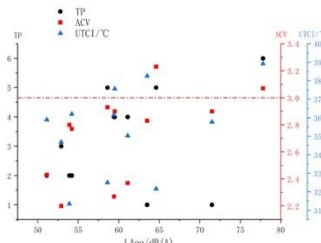

**Figure 8.** The values of ACV under different conditions of the thermal-acoustic environment.

Figure 9 illustrates the APV values at 12 different sites. Regarding the impact of a sound environment on APV, values tend to decrease when TP is below 4. As for the effect of the thermal environment on APV, UTCI does not have a significant overall impact.

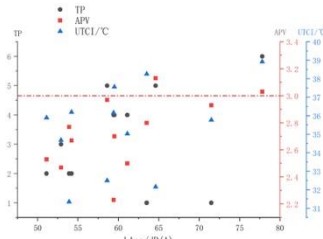

**Figure 9.** The values of APV under different conditions of the thermal-acoustic environment.

Figure 10 reveals the AHV values at the same 12 sites. Regarding the influence of a sound environment on AHV, values tend to increase when TP is below 4. In terms of the effect of the thermal environment on AHV, UTCI leads to an increase in deviation.

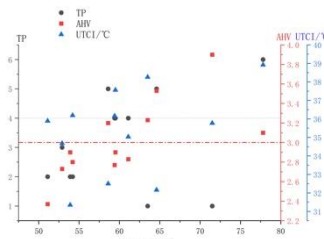

**Figure 10.** The values of AHV under different conditions of the thermal-acoustic environment.

Figure 11 displays the APlV values at the same 12 sites. Regarding the impact of a sound environment on APlV, SLV significantly increases as LAeq increases. When LAeq is over 53 dB(A), SLV exceeds the neutral level. As for the effect of the thermal environment on APlV, UTCI does not have a significant overall impact.

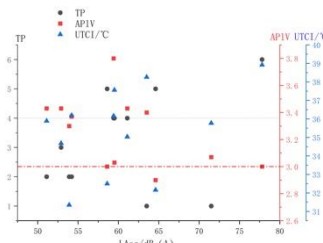

**Figure 11.** The values of APlV under different conditions of the thermal-acoustic environment.

Figure 12 exhibits the AFV values at the same 12 sites. Again, neither sound nor thermal environment has a significant overall impact on AFV.

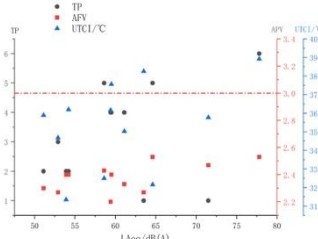

**Figure 12.** The values of AFV under different conditions of the thermal-acoustic environment.

Figure 13 illustrates the AIV values of 12 plots. Regarding the impact of the acoustic environment on AIV, AIV significantly increases with the increase of LAeq. Concerning the impact of the thermal environment on AIV, as the UTCI grows, AIV's deviation becomes larger.

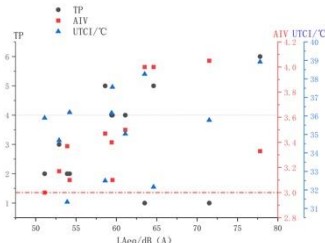

**Figure 13.** The values of AIV under different conditions of the thermal-acoustic environment.

Figure 14 depicts the AEV values of 12 plots. Concerning the impact of the acoustic environment on AEV, SLV significantly increases with the increase of LAeq. Regarding the impact of the thermal environment on AEV, as the UTCI grows, AEV's deviation becomes larger.

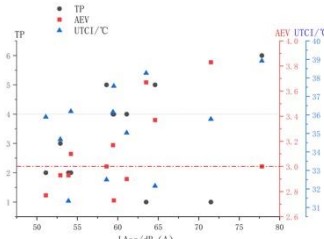

**Figure 14.** The values of AEV under different conditions of the thermal-acoustic environment.

Figure 15 shows the ADV values of 12 plots. Regarding the impact of the acoustic environment on ADV, LAeq has little effect. However, positive TP reduces ADV. Concerning the impact of the thermal environment on ADV, UTCI has no significant impact on the overall ADV.

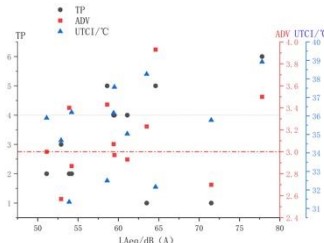

**Figure 15.** The values of ADV under different conditions of the thermal-acoustic environment.

Figure 16 displays the ACoV values for 12 plots. Regarding the impact of the acoustic environment on ACoV, LAeq has a negligible effect. However, positive TP reduces ACoV. Regarding the influence of the thermal environment on ACoV, an increase in UTCI results in higher ACoV deviations.

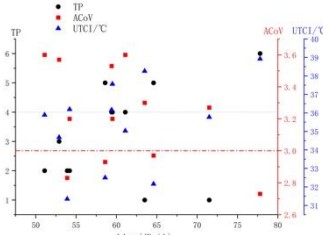

**Figure 16.** The values of ACoV under different conditions of the thermal-acoustic environment.

Figure 17 exhibits the OAV values for the same 12 plots. In terms of the effect of the acoustic environment on OAV, LAeq has no significant effect. Nevertheless, positive TP

increased OAV. Regarding the effect of the thermal environment on OAV, UTCI has little overall impact on OAV.

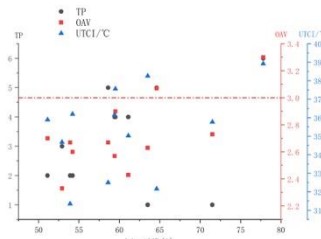

**Figure 17.** The values of OAV under different conditions of the thermal-acoustic environment.

### 3.4. Comprehensive Influence of Thermal Acoustic Environment on the Overall Environmental Assessment

To ensure consistency of the results and avoid misleading situations, the inversion of rating was implemented for OCV and OSV parameters. The impact of the thermal environment on overall environmental assessment includes three indicators: OCV, OAnV, and OSV. Table 8 presents the significance of these indicators under main effects and interaction effects (OCV, OAnV, OSV). The main effect of UTCI is significant on OCV ($p < 0.05$), OAnV ($p < 0.01$) and OSV ($p < 0.05$). The main impact of LAeq is significant on OAnV ($p < 0.05$). Moreover, the interaction effect of UTCI × LAeq is significant on OAnV ($p < 0.05$) and OSV ($p < 0.1$). Finally, TP significantly affects all three overall indicators ($p < 0.01$).

**Table 8.** The significance of the indicators under the main effect and interaction (OCV, OAnV and OSV).

| Subjective Evaluation | UTCI | LAeq | UTCI × LAeq | TP |
|:---:|:---:|:---:|:---:|:---:|
| OCV | **0.015** | 0.252 | 0.158 | **0** |
| OAnV | **0.004** | **0.026** | **0.018** | **0** |
| OSV | **0.02** | 0.11 | **0.086** | **0.003** |

Note: Bold indicates significant correlation within 0.1.

Figure 18 illustrates the OCV values under various thermal and acoustic conditions. Regarding the influence of the thermal environment on OCV, a higher UTCI value corresponds to an overall upward trend in OCV. This escalation in discomfort is evident. Regarding the impact of the acoustic environment on OCV, LAeq does not appear to have a significant effect on the overall OCV trend. However, when UTCI is less than 34 or greater than 37, TP impacts OCV, leading to a greater deviation. Moreover, when UTCI falls between 34 and 37, TP ≤ 4 tends to have a lower score for OCV.

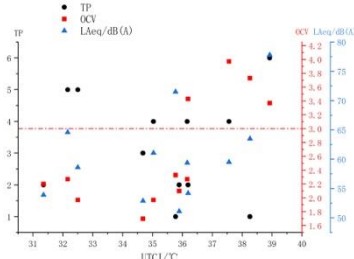

**Figure 18.** The values of OCV under different conditions of the thermal-acoustic environment.

Figure 19 presents the OAnV values under different thermal and acoustic conditions. Regarding the impact of the thermal environment on OAnV, as UTCI increases, OAnV demonstrates a significant upward trend. As for the influence of the acoustic environment on OAnV, higher values of LAeq and TP correspond to an overall increase in OAnV.

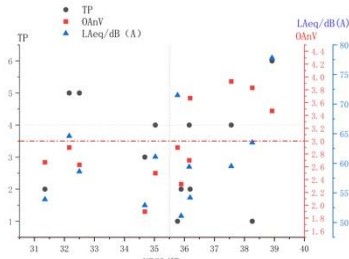

**Figure 19.** The values of OAnV under different conditions of the thermal-acoustic environment.

Figure 20 illustrates the OSV values under various thermal and acoustic environmental conditions. Although, regarding the impact of the thermal environment on OSV, there is a significant increase in OSV with an increase in UTCI, dissatisfaction with the manifestation is mounting. As for the impact of the acoustic environment on OSV, LAeq has little effect on OSV overall. However, when UTCI is less than 34 or greater than 37, TP can affect OSV and increase the deviation.

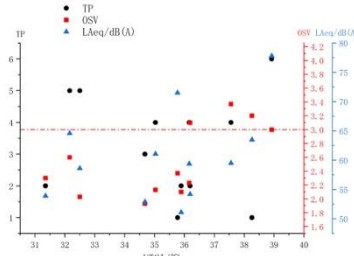

**Figure 20.** The values of OSV under different conditions of the thermal-acoustic environment.

## 4. Discussion

### 4.1. Interactions of the Thermal and Acoustic Environment

Comfort is a significant aspect in the design of urban parks as it influences the behavior of visitors and the comfort of the park environment. Varying environmental factors, especially the thermal and acoustic environment, can improve the park space experience. This study investigates the impact of outdoor thermal and acoustic environment interactions and related indicators in 12 sample locations within a city park (Xihu Park, Fuzhou City).

(1) The findings of this study indicate that the thermal soundscape impacts the SLV, AHV, AIV, AEV evaluation of urban parks. The results indicate that people perceive higher soundscape quality in comfortable and cool environments during summer. The effect of LAeq on SLV, AHV, AIV, AEV depends on the different UTCI levels. When UTCI is below 36, SLV is concentrated near a neutral level. However, when UTCI exceeds 36, the diversity of SLV, AHV, AIV, AEV significantly increases. Moreover, research has found that positive sound sources, such as wind blowing through leaves, water, and birdsong, can significantly improve the overall rating of sound evaluation. This can be explained by the fact that this diverse range of sounds can make people's auditory systems more active and sensitive [67]. Additionally, these natural soundscape sources stem from comfortable environments. Extensive empirical research suggests that the auditory stimuli individuals receive within natural surroundings promote greater physical and mental health and comfort compared to indoor environments. From a physiological perspective, natural sounds lower stress hormone levels, improve sleep quality, strengthen immune system functionality, and positively impact emotions and cognition [68]. Hearing natural sounds also enhances feelings of comfort and tranquility [69]. Therefore, hearing natural sounds can make people feel more comfortable and peaceful. Increasing vegetation cover and regulating environmental temperature in urban parks can significantly enhance acoustic quality. The urban heat island effect and other modern urban environmental issues have worsened the acoustic environment in urban parks due to noise pollution

and temperature rise. Reasonable vegetation arrangements, such as vegetation and tree configuration, can reduce the thermal resistance caused by building surface pavement and other hard surfaces, thus helping to alleviate the impact of temperature rise. In addition, with economic development and technological advancements, the artificial intervention has become a new method for adjusting the thermal environment of urban parks, such as using artificial wetting and increasing reflectivity. Furthermore, natural elements, especially trees, in the urban park environment create a variety of positive soundscapes, making the environment more pleasant [70,71]. These research findings are similar to previous studies [35,70,72–74]. During hot seasons, low-level thermal sensation (environmental cooling), high-level thermal comfort, and positive sound sources significantly enhance soundscape quality. Therefore, these research results can provide important reference information for urban park planning and construction.

(2) In this study, it was found that although there was no interaction between UTCI and LAeq in terms of heat evaluation, positive sound sources (TP) had a significant impact on TSV, TCV and OTV. Within the inner area of the park, vegetation was able to shield noise pollution, resulting in a relatively stable range of LAeq readings. Thus, the type of sound source in the park became the primary factor affecting heat evaluation, a finding also supported by other researchers [70,75,76]. Particularly during hot seasons, positive sound sources were effective in reducing TSV and increasing TCV and OTV. The type of sound source plays a significant role in regulating the sensory experience of humans in urban park environments. One can argue that positive acoustic environments, with their gradual and soft sound characteristics, aid in relaxation, easing the pressure of the heat on the body, and enhancing a cool and pleasurable feeling, ultimately improving the evaluation of thermal comfort and overall thermal environment. For instance, in an outdoor recreational area, the soothing and tranquil natural sounds, such as birds chirping, cicadas, or rustling leaves, enable people to withstand the discomfort of heat waves, thereby improving thermal comfort and overall evaluation of the thermal environment. Additionally, in urban park environments, it is vital to consider the ratio of acoustic landscapes to buildings, among other factors, to achieve the gradual variation of sound and ensure smooth sound transmission throughout the park [77]. This highlights the need to control and manage sound sources in the park's inner area to reduce visitor perceptions of heat during hot seasons. Furthermore, with increasing urbanization, noise pollution in urban parks is expected to worsen. Thus, improving the acoustic environment within parks will become an important task for park management in the future.

(3) We conducted an analysis of the overall environmental assessment, focusing on the interaction between UTCI and LAeq, and discovered a significant positive correlation between this interaction and OAnV and OSV. Specifically, as the deviation from the neutral temperature sound level increased, the overall annoyance votes also increased. This result is consistent with previous studies [75,78], indicating that in parks, the overall sense of annoyance and satisfaction of the human body are simultaneously affected by temperature and sound pressure. This is mainly because the human body is a highly sensitive biological organism, and various factors in the environment, such as sound, light, smell, and temperature, can stimulate the human nervous system and body. In a park, temperature and sound pressure changes can cause varying degrees of physiological and psychological changes in the human body, thereby affecting people's moods and emotions [79]. In terms of the actual environment, we also found other details. For example, in hot weather during the summer, more people tend to gather in the shaded and comfortable areas of the park. When noise levels reach a certain point, people feel even more annoyed, ultimately affecting the park's overall satisfaction level [80]. Based on these findings, we recommend implementing measures to reduce the impact of environmental factors on the human body when designing and managing parks. For instance, incorporating some shading facilities [81] to mitigate the temperature impact during summer and regulating noise sources like grass cutting [82] and traffic can significantly enhance the park's overall satisfaction and service quality.

*4.2. Shortcomings and Prospects*

There are certain limitations to this study. Firstly, it was conducted within a limited time frame (summer 2022). Thus, its scope and applicability may be confined to warm environmental conditions. Secondly, the influence of other relevant variables, such as the SVF index, may contribute towards the analysis and accuracy of results. Thirdly, in future research, including other effective factors like soil moisture content could enhance the accuracy of conclusions concerning plant community spatial partitioning. Fourthly, regarding factors influencing shade provision by trees, the correlation between the cooling effect and thermal comfort may be more significant than acoustic comfort and can be explored in future analyses. Fifthly, although our experiments recruited healthy subjects, there were still some atypical groups among the tourist population, such as those who suffered from depression or hearing impairment. These groups perceive the environment differently from healthy individuals. Therefore, future research should focus on exploring the influence of the acoustic and thermal environment on the perception of atypical groups.

Lastly, considering their unique ecological and environmental characteristics and the socioeconomic and cultural differences among users, conducting more research in different climatic locations is necessary.

**5. Conclusions**

This study utilized questionnaire surveys and monitoring of distinct thermal and acoustic environments to analyze whether these environments impact subjective evaluation and attempted to determine the effects of thermal, and acoustic environments on thermal, acoustic, and overall environmental evaluations.

(1) Regarding thermal evaluation, the results showed that the thermal environment significantly impacted thermal sensation, thermal comfort, thermal acception and overall thermal vote. The thermal sensation increased significantly as the temperature rose, while thermal comfort, thermal acception and overall thermal vote declined significantly. Regarding acoustic environments, since the park is located in a low-noise environment with minimal noise fluctuation, LAeq had no significant impact on the thermal environment; its impact was only manifest in the TP's impact on thermal sensation, thermal comfort and overall thermal vote. Amongst the six major sound sources (water, bird song, rustling leaves, broadcasting music, conversations, and construction noise), those ranking higher positively impacted thermal evaluation. However, there was no interaction between UTCI and LAeq on thermal assessment.

(2) Regarding acoustic evaluation, it can be observed that LAeq in the sound environment significantly impacts the subjective loudness, acoustic harshness, acoustic Intensity, and acoustic Excitability of the sampling site. As the Laeq value increases, the negative evaluation of Subjective loudness, acoustic Harshness, acoustic Intensity, and acoustic Excitability also increases. Furthermore, the acoustic Comfort, acoustic Preference, acoustic Disorder, acoustic Coordination, and overall acoustic Vote in the sampling site are also significantly affected by TP. The higher the rank of TP, the higher the positive evaluation of acoustic Comfort, acoustic Preference, acoustic Disorder, acoustic Coordination, and overall acoustic Vote. The thermal environment also affects acoustic Harshness, acoustic Intensity, acoustic Excitability, and acoustic Coordination. As the temperature rises, the corresponding indicators tend to have negative evaluations. Moreover, the interaction effect between UTCI and Laeq significantly impacts Subjective loudness, acoustic Harshness, acoustic Intensity, and acoustic Excitability. As UTCI and LAeq values increase, they will move away from the neutral level.

(3) In terms of overall environmental assessment, the hot environment significantly influences the overall comfort vote, overall Annoyance vote and overall satisfaction vote in all sample sites. As the temperature rises, there is a trend of negative evaluation in the overall comfort vote, annoyance vote, and satisfaction vote. Regarding the sound environment, LAeq impacts the assessment of overall annoyance vote in the sample sites, and its evaluation increases in environments with higher sound pressure levels. TP significantly

influences the overall comfort vote, overall annoyance vote and overall satisfaction vote in all sample sites, and the more prominent rankings of TP result in more positive evaluations. In addition, the interaction between UTCI and LAeq also affects the evaluation of the overall annoyance vote and overall satisfaction vote, wherein as UTCI and LAeq increase, their evaluations tend to be negative and move away from the neutral level.

**Author Contributions:** Conceptualization, Y.C. and J.D.; methodology, Y.C., F.L. and X.L.; software, Y.C., J.L. (Jing Liu) and Z.C.; validation, Y.C., J.L. (Junyi Li) and K.S.; formal analysis, Y.C. and F.L.; investigation, Y.C., F.L., X.L., J.L. (Jing Liu), Z.C., K.S. and J.L. (Junyi Li); resources, Y.C.; data curation, Y.C.; writing—original draft preparation, Y.C.; writing—review and editing, Y.C. All authors have read and agreed to the published version of the manuscript.

**Funding:** This research was funded by (1) Unit: Ministry of Science and Technology of the People's Republic of China. Green Urbanization across China and Europe: Collaborative Research on Key Technological Advances in Urban Forests, funding number 2021YFE0193200; (2) Unit: Ministry of Science and Technology of the People's Republic of China. Horizon 2020 strategic plan: CLEARING HOUSE-Collaborative Learning in Research, Information sharing, and Governance on How Urban tree-based solutions support Sino-European urban futures, funding number 821242.

**Data Availability Statement:** The data used to support the findings of this study are available from the corresponding author upon request.

**Conflicts of Interest:** The authors declare no conflict of interest.

## Appendix A

**Table A1.** Basic information of subjects.

| No. | Gender | Age | Time Living in Fuzhou | No. | Gender | Age | Time Living in Fuzhou |
|---|---|---|---|---|---|---|---|
| 01 | Male | 31~40 | 16 | 16 | Male | 18~24 | 6 |
| 02 | Female | 31~40 | 12 | 17 | Female | 18~24 | 7 |
| 03 | Male | 25~30 | Less than 1 year | 18 | Female | 18~24 | Less than 1 year |
| 04 | Female | 25~30 | 10 | 19 | Male | 25~30 | Local |
| 05 | Male | 31~40 | 11 | 20 | Female | 25~30 | 12 |
| 06 | Male | 31~40 | 12 | 21 | Female | 31~40 | Less than 1 year |
| 07 | Male | 25~30 | Less than 1 year | 22 | Male | 18~24 | Local |
| 08 | Male | 25~30 | 12 | 23 | Male | 25~30 | 7 |
| 09 | Male | 25~30 | 2 | 24 | Female | 18~24 | Local |
| 10 | Male | 31~40 | Local | 25 | Female | 25~30 | Local |
| 11 | Male | 25~30 | 8 | 26 | Female | 25~30 | 12 |
| 12 | Male | 18~24 | 5 | 27 | Male | 25~30 | 7 |
| 13 | Female | 18~24 | 5 | 28 | Male | 25~30 | Less than 1 year |
| 14 | Male | 25~30 | 9 | 29 | Female | 18~24 | Local |
| 15 | Female | 25~30 | 7 | 30 | Female | 18~24 | Local |

**Table A2.** Subjective evaluations of the questionnaire surveys (TSV, TCV, TAV, OTV).

| Please Tick off One Response Alternatives for the Surrounding Thermal Environment | 1 | 2 | 3 | 4 | 5 |
|---|---|---|---|---|---|
| Thermal Sensation | ☐ Cold | ☐ Cool | ☐ Neutral | ☐ warm | ☐ Hot |
| Thermal Comfort | ☐ Uncomfortable | ☐ Slightly Uncomfortable | ☐ Neutral | ☐ Slightly Comfortable | ☐ Comfortable |
| Thermal Acceptance | ☐ Unacceptable | ☐ Slightly unacceptable | ☐ Neutral | ☐ Slightly acceptable | ☐ Acceptable |
| Overall Thermal | ☐ Bad | ☐ Slightly bad | ☐ Neutral | ☐ Slightly good | ☐ Good |

**Table A3.** Subjective evaluations of the questionnaire surveys (SLV, ACV, APV, AHV, APlV, AFV, AIV, AEV, ADV, ACoV, OAV).

| Please Tick off the 11 Options That Match Your Feelings about the Surrounding Sound Environment | 1 | 2 | 3 | 4 | 5 |
|---|---|---|---|---|---|
| Subjective Loudness | ☐ Quiet | ☐ Slightly quiet | ☐ Neutral | ☐ Slightly Loud | ☐ Loud |
| Acoustic Comfort | ☐ Uncomfortable | ☐ Slightly Uncomfortable | ☐ Neutral | ☐ Slightly Comfortable | ☐ Comfortable |
| Acoustic Preference | ☐ Unlike | ☐ Slightly unlike | ☐ Neutral | ☐ Slightly like | ☐ Like |
| Acoustic Harshness | ☐ Gentle | ☐ Slightly Gentle | ☐ Neutral | ☐ Slightly harsh | ☐ Harsh |
| Acoustic Pleasure | ☐ Unhappy | ☐ Slightly Unhappy | ☐ Neutral | ☐ Slightly happy | ☐ Happy |
| Acoustic Familiarity | ☐ Strange | ☐ Slightly Strange | ☐ Neutral | ☐ Slightly familiar | ☐ Familiar |
| Acoustic Intensity | ☐ Weak | ☐ Slightly weak | ☐ Neutral | ☐ Slightly strong | ☐ Strong |
| Acoustic Excitability | ☐ Calm | ☐ Slightly Calm | ☐ Neutral | ☐ Slightly excited | ☐ Excited |
| Acoustic Disorder | ☐ Monotonous | ☐ Slightly monotonous | ☐ Neutral | ☐ Slightly disordered | ☐ Disordered |
| Acoustic Coordination | ☐ Incoordinate | ☐ Slightly incoordinate | ☐ Neutral | ☐ Slightly coordinate | ☐ Coordinate |
| Overall Acoustic | ☐ Bad | ☐ Slightly bad | ☐ Neutral | ☐ Slightly good | ☐ Good |

**Table A4.** Subjective evaluations of the questionnaire surveys (OCV, OAnV, OSV).

| Overall, How Would You Describe the Present Surroundings Acoustic-Thermal Environment? | 1 | 2 | 3 | 4 | 5 |
|---|---|---|---|---|---|
| Overall Comfort | ☐ Uncomfortable | ☐ Slightly Uncomfortable | ☐ Neutral | ☐ Slightly Comfortable | ☐ Comfortable |
| Overall Annoyance | ☐ Peaceful | ☐ Slightly Peaceful | ☐ Neutral | ☐ Slightly annoying | ☐ Annoying |
| Overall Satisfaction | ☐ Bad | ☐ Slightly bad | ☐ Neutral | ☐ Slightly good | ☐ Good |

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
