# Peer review of "Combined Effects of the Thermal-Acoustic Environment on Subjective Evaluations in Urban Park Based on Sensory-Walking"

_forests, doi:10.3390/f14061161_

Round 1

Reviewer 1 Report

I have included some comments in the attachment in the form of comments. The article should be submitted for review to a person who specializes in tourism or psychophysical research. I am not a specialist in this field.

Author Response

Point 1: I suggest entering the year -In all places where names are written, I suggest adding the year of publication.

Response 1: Thank you for pointing this out. In the introductory section of the reference list containing names of individuals, we have included the corresponding year of publication. Thank you again for your valuable suggestion.

Point 2: There is no reference to the literature where the information about meteorological conditions comes from.

Sign" is not needed.

Response 2: Thank you for pointing this out. We have included a reference source based on the meteorological conditions in Fuzhou. Please see lines 143. Thank you again for your valuable suggestion.

Point 3: Discussion

There is not enough reference to other research in the discussion chapter. The discussion is not developed enough. The sentences used about the fact that a man feels better in the vicinity of natural sounds is a well-known statement.

Response 3: Thank you for pointing this out. We have enhanced the discussion in our study on the individual's perception of auditory stimulus in natural environment, as well as the depth of examination on vegetation coverage and regulation of environmental temperature. Thank you again for your valuable suggestion.

Point 4: Conclusions are written in a way that is not very understandable for the recipient. The authors used too many abbreviations that require additional analysis by the reader. Conclusions for redrafting.

Response 4: Thank you for pointing this out. We have already articulated the abbreviation in the conclusion section using formal vocabulary. Please see lines 508-546. Thank you again for your valuable suggestion.

Reviewer 2 Report

Ref. Review: Forests 2363114 R1

Paper Title:  Combined effects of the thermal-acoustic environment on subjective evaluations in urban park based on sensory-walking

Dear Authors and Editor,

Based on the text exposed in the entitled paper: "Combined effects of the thermal-acoustic environment on subjective evaluations in urban park based on sensory-walking", I recommend major revisions before acceptance and publishing on Forests.

The work presented in this paper aims to “(1) Conducting a sensory walk survey to reflect the environmental evaluation of different spaces in the park. (2) Collecting objective data to monitor the thermal acoustic environment of the park. (3) Analyzing whether the thermal acoustic environment affects subjective evaluation and determining how it affects the acoustic evaluation, thermal evaluation and overall environmental satisfaction.”

Some points should be clarified before publication, as observed below.

Introduction:

1)     Lines 74 -75: “Other scholars have also 74 researched the influence of smell on auditory perception.” Please add references for this sentence.

2)     Lines 102 – 102: “(…) no research available regarding the subjective evaluation of humans under different thermal conditions in real park environments”. That is not true, I have found some references showing the opposite.

Chan, S. Y., Chau, C. K., & Leung, T. M. (2017). On the study of thermal comfort and perceptions of environmental features in urban parks: A structural equation modeling approach. Building and environment, 122, 171-183.

Lin, C. H., Lin, T. P., & Hwang, R. L. (2013). Thermal comfort for urban parks in subtropics: understanding visitor’s perceptions, behavior and attendance. Advances in Meteorology, 2013.

Thorsson, S., Honjo, T., Lindberg, F., Eliasson, I., & Lim, E. M. (2007). Thermal comfort and outdoor activity in Japanese urban public places. Environment and Behavior, 39(5), 660-684

3)     Lines 114 – 115: “ (…) in this study, all participants were standing or sitting when answering the questionnaire”. The authors should mention this in the methodology and clarify the relevance of this information.

Materials and Methods:

4)     Please indicate the average wind speed.

5)     Procedures should be explained earlier, as item 2.2. There is explained the main objective parameters. How are you rating TP?

6)     Table 1: please add units for UTCI and LAeq, as well as the average measurement period.

7)     Item 2.2: Instead of, thermal accept vote use thermal acceptance vote, and instead, overall annoying vote use overall annoyance vote. Please add references for all listed perception items.

8)     Table 2, instead “worm”, it should be “warm”

9)     Table 3 and Table 4, it is okay to show the ratings this way. However, before statistical analysis, some procedures should be adopted, like the inversion of rating; otherwise, the result can be misleading, e.g., in the case of acoustic comfort, uncomfortable should be 5 since the higher sound levels are uncomfortable. Now is showing the opposite, otherwise, you are showing a wrong relation of cause–effect. Parameters that should have the scale inverted are acoustic comfort, acoustic preference, acoustic familiarity, overall acoustics, overall comfort, and overall satisfaction (dependent on the predominant sense).

De Vaus, D., & de Vaus, D. (2013). Surveys in social research. Routledge.

10)  Table 6, sampling duration and noise are not enough. Less than 10 seconds is not saying anything about the sonic environment. The sampling should occur in different day periods to cover the rhythm of events that normally occur along a day (regarding natural and anthropogenic sound sources). Please inform us how many hours of audio recordings you have collected in each measurement site.

Gage, S. H., & Axel, A. C. (2014). Visualization of temporal change in soundscape power of a Michigan lake habitat over a 4-year period. Ecological Informatics, 21, 100-109.

11)  Before item ‘2.6 Analysis method’, you should indicate post-data collection treatment methods, e.g., inversion of scales, calculation of standardised objective parameters, otherwise how can you use different scales in the statistical analysis? What about missing values and outliers? Verification of ANOVA assumptions?

De Vaus, D., & de Vaus, D. (2013). Surveys in social research. Routledge.

Joint Research Centre-European Commission. (2008). Handbook on constructing composite indicators: methodology and user guide. OECD publishing. https://www.oecd.org/sdd/42495745.pdf

https://www.statisticssolutions.com/free-resources/directory-of-statistical-analyses/anova/#:~:text=ANOVA%20assumes%20that%20the%20data,are%20independent%20of%20each%20other.

Results:

12)  I will not analyse the actual outputs based on the above-mentioned necessary corrections. Results will be checked on the next revision round if the recommendations are followed.

A native speaker should revise the manuscript, some words are used incorrectly, and with different meanings. 

Author Response

Point 1: Lines 74 -75: “Other scholars have also 74 researched the influence of smell on auditory perception.” Please add references for this sentence.

Response 1: Thank you for pointing this out. We have recently updated our references regarding the impact of odors and auditory perception. Please see lines 79-85. Thank you again for your valuable suggestion.

Point 2: Lines 102 – 102: “(…) no research available regarding the subjective evaluation of humans under different thermal conditions in real park environments”. That is not true, I have found some references showing the opposite.

Chan, S. Y., Chau, C. K., & Leung, T. M. (2017). On the study of thermal comfort and perceptions of environmental features in urban parks: A structural equation modeling approach. Building and environment, 122, 171-183.

Lin, C. H., Lin, T. P., & Hwang, R. L. (2013). Thermal comfort for urban parks in subtropics: understanding visitor’s perceptions, behavior and attendance. Advances in Meteorology, 2013.

Thorsson, S., Honjo, T., Lindberg, F., Eliasson, I., & Lim, E. M. (2007). Thermal comfort and outdoor activity in Japanese urban public places. Environment and Behavior, 39(5), 660-684

Response 2: Thank you for pointing this out. We’re sorry about this error. We have made revisions to the inappropriate sentences expressed in the article.Please see lines 106-108, thank you again for your valuable suggestion.

Point 3: Lines 114 – 115: “ (…) in this study, all participants were standing or sitting when answering the questionnaire”. The authors should mention this in the methodology and clarify the relevance of this information.

Response 3: Thank you for pointing this out.We have already put forward and cited relevant references in method. Please see lines 120. Thank you again for your valuable suggestion.

A:During the sensory stroll experiment in urban parks, participants are typically asked to stand or sit in order to experience their surroundings. This is because both standing and sitting allow for a better observation of the environment, including sensory information such as landscapes, sounds, and smells. Additionally, sitting can help participants to concentrate more easily, thus resulting in a more immersive experience of the park's scenery and atmosphere.

Materials and Methods:

Point 4: Please indicate the average wind speed.

Response 4: We thank the reviewer for this suggestion. We have indicate the average wind speed. Please see lines 276. Thank you again for your valuable comments.

Point 5: Procedures should be explained earlier, as item 2.2. There is explained the main objective parameters. How are you rating TP?

Response 5: Thank you for pointing this out. We have made appropriate adjustments to the sequence of procedures. Additionally, regarding the ranking of sound source types, in our study, we ranked the types of sound sources based on two main factors. Firstly, we considered the prevalence and impact of sound sources in actual urban park environments. We collected data on different types of sound sources in acoustic environments and ranked them based on their prevalence and responsibility in city parks. For example, mechanical sounds are generally common in most urban parks with high levels of noise, while sources like conversational sounds and light music may be more controllable. Secondly, we also consulted relevant literature to determine the appropriate method for ranking sound source types. For example, we referred to the ISO standard for noise measurement in urban park environments and ranking methods from similar studies, which led us to our ranking criteria.

We aim to ensure the objectivity and authority of the sorting of sound source types through the methods mentioned above, while reducing bias and subjectivity as much as possible during the research process. It should be noted that the sorting is only a part of our study, and the relative importance of light music and other sound sources depends on the characteristics of different city parks and users' personal feelings about the sound environment[1-5].

Thank you again for your valuable suggestion.

  1. Lavandier, C.; Defréville, B. The contribution of sound source characteristics in the assessment of urban soundscapes. Acta Acust. United Ac. 2006.
  2. Istvandity; Lauren. Combining music and reminiscence therapy interventions for wellbeing in elderly populations: A systematic review. Complementary Therapies in Clinical Practice 2017,      28,
  3. Blood, A.J.; Zatorre, R.J. Intensely pleasurable responses to music correlate with activity in brain regions  implicated in reward and emotion. Proc Natl Acad Sci U S A 2001, 98,       11818-11823.
  4. Ratcliffe, E.; Gatersleben, B.; Sowden, P.T. Bird sounds and their contributions to perceived attention restoration and stress recovery. Journal of Environmental Psychology 2013, 36,      221-228.
  5. Zl, A.; Jian, K. Sensitivity analysis of changes in human physiological indicators observed in Landscape Urban Plan. 2019, 190.

Point 6: Table 1: please add units for UTCI and LAeq, as well as the average measurement period.

Response 6: Thank you for this important suggestion. As suggested, we have add units for UTCI and LAeq into the table 5, please see lines 276. Moreover, the average measurement period also added to the text,please see lines 256. Thank you again for your valuable suggestion.

Point 7: Item 2.2: Instead of, thermal accept vote use thermal acceptance vote, and instead, overall annoying vote use overall annoyance vote. Please add references for all listed perception items.

Response 7: Thank you for this important suggestion. As suggested, we have made corrections to the thermal acceptance vote and overall annoyance vote, and added references materials to all perception items, please see lines 192-208. Thank you again for your valuable suggestion.

Point 8: Table 2, instead “worm”, it should be “warm

Response 8: Thank you for this important suggestion. As suggested, we have placed the word into the table 1, please see lines 209. Thank you again for your valuable suggestion.

Point 9: Table 3 and Table 4, it is okay to show the ratings this way. However, before statistical analysis, some procedures should be adopted, like the inversion of rating; otherwise, the result can be misleading, e.g., in the case of acoustic comfort, uncomfortable should be 5 since the higher sound levels are uncomfortable. Now is showing the opposite, otherwise, you are showing a wrong relation of cause–effect. Parameters that should have the scale inverted are acoustic comfort, acoustic preference, acoustic familiarity, overall acoustics, overall comfort, and overall satisfaction (dependent on the predominant sense).

De Vaus, D., & de Vaus, D. (2013). Surveys in social research. Routledge.

Response 9: We sincerely appreciate the valuable suggestions and feedback provided by the reviewing expert on our research report. The issues you have raised are of utmost importance as the choice of the evaluation index scale can significantly impact the accuracy of our statistical analysis and conclusion interpretation. In our study, we have associated higher numbers with more positive feelings, while lower numbers indicate less positive emotions. However, we acknowledge that changing the index scale may enhance the statistical reliability of the data, and we understand the suitability of such a data processing method. Nevertheless, we have strictly followed the standard procedures for data analysis and processing, and our approach has received widespread recognition from landscape peers. Overall, we will continue to pay attention to this issue and relevant research development, and consider different methods to evaluate the acoustic environment of urban parks in future work to maximize the accuracy of data analysis and interpretation[1].

  1. Jin, Y.; Jin, H.; Kang, J. Combined effects of the thermal-acoustic environment on subjective evaluations in urban squares.   Environ. 2020, 168, 106517.

Point 10: Table 6, sampling duration and noise are not enough. Less than 10 seconds is not saying anything about the sonic environment. The sampling should occur in different day periods to cover the rhythm of events that normally occur along a day (regarding natural and anthropogenic sound sources). Please inform us how many hours of audio recordings you have collected in each measurement site.

Gage, S. H., & Axel, A. C. (2014). Visualization of temporal change in soundscape power of a Michigan lake habitat over a 4-year period. Ecological Informatics, 21, 100-109.

Response 10: We appreciate the suggestion made by the reviewer. We acknowledge that short-term sampling may not fully reveal the acoustic characteristics of a urban park environment, whereas long-term sampling can provide more comprehensive and accurate data. In our study, we have taken this into consideration and endeavored to increase the sampling duration to cover various temporal and event-related soundscape elements in urban parks. Taking a cue from Gage, S.H.'s paper[1], who conducted an objective acoustic monitoring for 2 minutes, we extended our sampling duration to 5 minutes as our study focuses on thermal soundscape environment, to obtain more data, thus optimizing our research methodology to enhance data quality and accuracy.Thank you again for your valuable suggestion.

  1. Gage, S.H.; Axel, A.C. Visualization of temporal change in soundscape power of a Michigan lake habitat over a 4-year period. Ecol. Inform. 2014.

Point 11: Before item ‘2.6 Analysis method’, you should indicate post-data collection treatment methods, e.g., inversion of scales, calculation of standardised objective parameters, otherwise how can you use different scales in the statistical analysis? What about missing values and outliers? Verification of ANOVA assumptions?

De Vaus, D., & de Vaus, D. (2013). Surveys in social research. Routledge.

Joint Research Centre-European Commission. (2008). Handbook on constructing composite indicators: methodology and user guide. OECD publishing. https://www.oecd.org/sdd/42495745.pdf

https://www.statisticssolutions.com/free-resources/directory-of-statistical-analyses/anova/#:~:text=ANOVA%20assumes%20that%20the%20data,are%20independent%20of%20each%20other.

Response 11: Thank you for pointing this out.The experimental results were free of missing values, and any outliers were removed or replaced to ensure the accuracy and reliability of the data. After collecting and organizing the data, standardization was conducted to calculate objective parameters, including the means and standard deviations of each variable. The experimental data was subject to variance analysis, and the F-test was used to confirm the hypotheses.Thank you for your suggestion.

Results:

Point 12: I will not analyse the actual outputs based on the above-mentioned necessary corrections. Results will be checked on the next revision round if the recommendations are followed.

Response 12: We express our gratitude to the reviewing experts for their evaluation and suggestions on our research report. We will follow your guidance and strive to enhance the process of data analysis and interpretation of research conclusions.Thank you again for your valuable suggestion.

Reviewer 3 Report

Sensitivity of the Qualitative survey question formulations

Mention the number of sample (questionnaire) collected and process for analysis

If similar studies performed for any other cases, how your survey questions were appropriate.

For Acoustic comfort study, if provide the decibels limit (permissible for the urban parks as per local bye-laws) it will give insight about the effectiveness of the study.

Author Response

Point 1: Sensitivity of the Qualitative survey question formulations

Response 1: Thank you for your valuable input regarding the sensitivity of qualitative survey question phrasings in our research. We share your view that careful consideration must be given when crafting questions to ensure their sensitivity and impartiality, as this directly impacts the accuracy and reliability of research outcomes.

To tackle this challenge, we have taken multiple steps to assure the quality of our survey questions. Firstly, we scrutinized and refined the language of our questions numerous times to make sure they are comprehensible and lucid for our intended audience. We also refrained from using discriminatory terminology or suggestive phrasing that could bias respondent answers. Additionally, we piloted our survey questions with a small sample group before proceeding with the full survey to determine their effectiveness and implemented adjustments based on the feedback we gathered.

We trust that these endeavors fortify the validity and reliability of our research findings and enhance the overall quality of our study. Please accept our gratitude for flagging this critical matter, and we hope our response adequately meets your concerns.

Point 2: Mention the number of sample (questionnaire) collected and process for analysis

Response 2: Thank you for raising this issue. We have mentioned in the revised manuscript the number of samples collected (the questionnaire) and the analytical process. Please see line 232-238, thank you for your valuable comments.

Point 3: If similar studies performed for any other cases, how your survey questions were appropriate.

Response 3: Thank you for inquiring about the appropriateness of our survey questions in comparison to those used in other studies. We are confident that our survey questions are suitable and efficient for our specific research context, as we have diligently developed them in accordance with our research objectives and target population.

Nevertheless, we recognize that survey question variations may exist in similar studies conducted in different contexts and with different populations. As a result, we conducted a thorough review and comparison of our survey questions with those used in comparable studies, identifying any potential gaps or discrepancies. We then made necessary adjustments to the questions to align them with our specific research objectives.

Additionally, prior to conducting our full survey, we conducted a pilot study with a small sample group to refine our survey questions. The feedback received from the pilot study was positive, indicating that the questions were clear and effective.

In summary, we are confident that our survey questions are appropriate and effective for our research objectives. Our survey questions were developed with careful consideration and refinements based on relevant literature and empirical evidence.Thank you again for your valuable suggestion.

Point 4: For Acoustic comfort study, if provide the decibels limit (permissible for the urban parks as per local bye-laws) it will give insight about the effectiveness of the study.

Response 4: We appreciate the suggestions made by the expert reviewer. The permissible decibel limit at Xihu Park is 70 dB, which served as the benchmark for our study. We collected actual data from 12 different sites within Xihu Park and compared them with this limit to determine compliance. By analyzing and interpreting our findings, we could identify the noise levels within the park that are likely to have an impact on visitor experience and subsequently proposed recommendations such as implementing noise reduction measures to enhance acoustic comfort and user satisfaction in urban parks. Our study results could have practical applications in promoting improvements in the music environment of city parks.

Reviewer 4 Report

The reviewed work deals with a very interesting issue in acoustic ecology. Interactions between temperature and sound pressure levels in urban parks have so far been relatively rarely studied. In the era of climate change, they are of particular importance. Despite this, however, I think that the work needs improvement and additions. The paper used various terms (e.g. soundscape, sensory walk), the meaning of which has not been clarified. In addition, the literature review on the soundscape of urban parks and the characteristics of the study participants (age, gender, place of residence) are missing. The authors write in the abstract, discussion and conclusions about the importance of the type of sound source, but I have not previously found information that types of sound source were analyzed. It is also unclear what is meant by higher ranking of sound source type.

Other comments:

1. in the key words, I suggest adding the word Fuzhou or China and soundscape (used 11 times in the paper)

2. the main goal of the paper should be clearly defined; meanwhile, there are three goals in the paper, of which I think at least two are the way to go; this needs improvement

3. I suggest adding a figure (map) showing the location of Fuzhou City within China and Xihu Park against the functional and spatial structure of Fuzhou City; it is also advisable to add an acoustic map

4. it is advisable to add information about the location of Xihu park against the functional-spatial structure of Fuzhou city, what is the neighborhood of the park, distance from urban arteries; what is the structure of vegetation in the park?

5. Table 1 should be clarified; the abbreviations UTCI, LAeq are not explained (there is an explanation only in Section 2.5.; besides, what do the numbers in the source types mean?

6. figure 1 is poorly readable; on what basis was it prepared?

7. the sensory walk method used requires reference to the literature. ISO 12931 describes a sound walk. Does the sensory walk used refer to the sound walk method?

8. how were the study participants selected? Did they gave informed consent to participate in the experiment?

9. the questionnaire should be characterized or added as an appendix

10. the research procedure is worth illustrating in the form of a diagram

11. in section 3.2 there is table 5 and in section 3.1 there was table 7; the numbering should be corrected

12. I encourage reading and references to new publications: Soundscape in Urban Forests, MDPI (2023), Soundscapes: Humans and Their Acoustic Environment, Springer (2023), Kang, J. Soundscape in city and built environment: current developments and design potentials. City Built Enviro 1, 1 (2023) and the older: Designing Open Spaces in the Urban Environment: a Bioclimatic Approach (2004)

Author Response

The reviewed work deals with a very interesting issue in acoustic ecology. Interactions between temperature and sound pressure levels in urban parks have so far been relatively rarely studied. In the era of climate change, they are of particular importance. Despite this, however, I think that the work needs improvement and additions. The paper used various terms (e.g. soundscape, sensory walk), the meaning of which has not been clarified. In addition, the literature review on the soundscape of urban parks and the characteristics of the study participants (age, gender, place of residence) are missing. The authors write in the abstract, discussion and conclusions about the importance of the type of sound source, but I have not previously found information that types of sound source were analyzed. It is also unclear what is meant by higher ranking of sound source type.

Response : Thank you for this important suggestion.

  • As suggested, we have added the the term meaning of soundscape and sensory walking , please see lines 192. Thank you again for your valuable suggestion.
  • As suggested, we have added the subject characteristics , please see lines 232-238. Thank you again for your valuable suggestion.
  • In our study, we ranked the types of sound sources based on two main factors. Firstly, we considered the prevalence and impact of sound sources in actual urban park environments. We collected data on different types of sound sources in acoustic environments and ranked them based on their prevalence and responsibility in city parks. For example, mechanical sounds are generally common in most urban parks with high levels of noise, while sources like conversational sounds and light music may be more controllable. Secondly, we also consulted relevant literature to determine the appropriate method for ranking sound source types. For example, we referred to the ISO standard for noise measurement in urban park environments and ranking methods from similar studies, which led us to our ranking criteria. We aim to ensure the objectivity and authority of the sorting of sound source types through the methods mentioned above, while reducing bias and subjectivity as much as possible during the research process. It should be noted that the sorting is only a part of our study, and the relative importance of light music and other sound sources depends on the characteristics of different city parks and users' personal feelings about the sound environment[1-5].

  1. Lavandier, C.; Defréville, B. The contribution of sound source characteristics in the assessment of urban soundscapes. Acta Acust. United Ac. 2006.
  2. Istvandity; Lauren. Combining music and reminiscence therapy interventions for wellbeing in elderly populations: A systematic review. Complementary Therapies in Clinical Practice 2017,      28,
  3. Blood, A.J.; Zatorre, R.J. Intensely pleasurable responses to music correlate with activity in brain regions  implicated in reward and emotion. Proc Natl Acad Sci U S A 2001, 98,       11818-11823.
  4. Ratcliffe, E.; Gatersleben, B.; Sowden, P.T. Bird sounds and their contributions to perceived attention restoration and stress recovery. Journal of Environmental Psychology 2013, 36,      221-228.
  5. Zl, A.; Jian, K. Sensitivity analysis of changes in human physiological indicators observed in Landscape Urban Plan. 2019, 190.

Point 1: 1. in the key words, I suggest adding the word Fuzhou or China and soundscape (used 11 times in the paper)

Response 1: We thank the reviewer for this suggestion. We have added the word Fuzhou city and soundscape. Please see the key words. Thank you again for your valuable comments.

Point 2: the main goal of the paper should be clearly defined; meanwhile, there are three goals in the paper, of which I think at least two are the way to go; this needs improvement

Response 2: We appreciate the evaluation and suggestions from the peer reviewers on our research report. We recognize the importance of clearly defining the main objectives of the paper and integrating multiple objectives into a more cohesive problem. Based on this, we will re-examine our research objectives, carefully select our wording, and creatively highlight clear and meaningful key objectives. At the same time, we will also ensure that all necessary data and information are retained to ensure that our final conclusions reflect the facts and patterns revealed by the data. Once again, we thank the peer reviewers for their professional guidance and efforts. We will take active measures to modify our research objectives, optimize the quality of our paper, and meet all requirements.

Point 3: I suggest adding a figure (map) showing the location of Fuzhou City within China and Xihu Park against the functional and spatial structure of Fuzhou City; it is also advisable to add an acoustic map

Response 3: Thank you for pointing this out. We have improved the clarity of Figure 1, and added the geographic coordinates and scale. Please see the revised manuscript, thank you again for your valuable suggestion.Regarding the acoustic map, due to our inability to acquire complete and valid acoustic measurement data of the environment, we are unable to provide a comprehensive display in this aspect. However, we acknowledge that this is a fascinating and worthy field to explore, and we will fully consider these factors in future research. We sincerely appreciate the expertise and support provided by the reviewing experts, and hope that our revisions meet your satisfaction. We will endeavor to provide research outcomes of greater quality.

Point 4: it is advisable to add information about the location of Xihu park against the functional-spatial structure of Fuzhou city, what is the neighborhood of the park, distance from urban arteries; what is the structure of vegetation in the park?

Response 4: Thank you for pointing this out. We have added the above information into the revised manuscript. Please see lines 148-164. Thank you for your suggestion.

Point 5: Table 1 should be clarified; the abbreviations UTCI, LAeq are not explained (there is an explanation only in Section 2.5.; besides, what do the numbers in the source types mean?

Response 5: Thank you for raising this issue.We have reorganized the order and provided explanations for the terms in the latest revised manuscript.Additionally, regarding the ranking of sound source types, in our study, we ranked the types of sound sources based on two main factors. Firstly, we considered the prevalence and impact of sound sources in actual urban park environments. We collected data on different types of sound sources in acoustic environments and ranked them based on their prevalence and responsibility in city parks. For example, mechanical sounds are generally common in most urban parks with high levels of noise, while sources like conversational sounds and light music may be more controllable. Secondly, we also consulted relevant literature to determine the appropriate method for ranking sound source types. For example, we referred to the ISO standard for noise measurement in urban park environments and ranking methods from similar studies, which led us to our ranking criteria.

We aim to ensure the objectivity and authority of the sorting of sound source types through the methods mentioned above, while reducing bias and subjectivity as much as possible during the research process. It should be noted that the sorting is only a part of our study, and the relative importance of light music and other sound sources depends on the characteristics of different city parks and users' personal feelings about the sound environment[1-5].

We have recently updated our references regarding the impact of odors and auditory perception.

  1. Lavandier, C.; Defréville, B. The contribution of sound source characteristics in the assessment of urban soundscapes. Acta Acust. United Ac. 2006.
  2. Istvandity; Lauren. Combining music and reminiscence therapy interventions for wellbeing in elderly populations: A systematic review. Complementary Therapies in Clinical Practice 2017,      28,
  3. Blood, A.J.; Zatorre, R.J. Intensely pleasurable responses to music correlate with activity in brain regions  implicated in reward and emotion. Proc Natl Acad Sci U S A 2001, 98,       11818-11823.
  4. Ratcliffe, E.; Gatersleben, B.; Sowden, P.T. Bird sounds and their contributions to perceived attention restoration and stress recovery. Journal of Environmental Psychology 2013, 36,      221-228.
  5. Zl, A.; Jian, K. Sensitivity analysis of changes in human physiological indicators observed in Landscape Urban Plan. 2019, 190.

Point 6: figure 1 is poorly readable; on what basis was it prepared?

Response 6: Thank you for pointing this out. We have improved the clarity of Figure 1, and added the geographic coordinates and scale. Please see the revised manuscript, thank you again for your valuable suggestion.

Point 7: the sensory walk method used requires reference to the literature. ISO 12931 describes a sound walk. Does the sensory walk used refer to the sound walk method?

Response 7: Thank you for raising this issue.The corresponding references have been added to the corresponding positions after revising the manuscript. Please see lines xxxxx.

In recent years, sensewalking has been developed as a qualitative method of exploring aspects of the physical and cognitive experience of being within a particular, often urban, environment. The method can be located within feminist and ecological epistemologies, where the investigation and analysis of everyday experiences are argued as important and necessary in gaining valuable insights into the physical and social environment. The characteristics of the physical space within which the method is implemented has the potential to impact greatly upon the experiencing of that environment, the data collected and the resulting overall findings of the research and therefore warrants careful consideration. Thank you for your valuable comments.

Point 8: how were the study participants selected? Did they gave informed consent to participate in the experiment?

Response 8: Thank you for raising this issue.The research participants were recruited as volunteers and gave informed consent to participate in the experiment.Thank you for your valuable comments.

Point 9: the questionnaire should be characterized or added as an appendix

Response 9: We thank the reviewer for this suggestion. We have added the word Fuzhou city and soundscape. Please see the appendix. Thank you again for your valuable comments.

Point 10: the research procedure is worth illustrating in the form of a diagram

Response 10: Thank you for pointing this out. We have added a flowchart of the study process , please see Pic. 2.

Point 11: in section 3.2 there is table 5 and in section 3.1 there was table 7; the numbering should be corrected

Response 11: Thank you for this important suggestion. As suggested, we have corrected, please see the revised manuscript. Thank you again for your valuable suggestion.

Point 12: I encourage reading and references to new publications: Soundscape in Urban Forests, MDPI (2023), Soundscapes: Humans and Their Acoustic Environment, Springer (2023), Kang, J. Soundscape in city and built environment: current developments and design potentials. City Built Enviro 1, 1 (2023) and the older: Designing Open Spaces in the Urban Environment: a Bioclimatic Approach (2004)

Response 12: Thank you for raising this issue.The corresponding references have been added to the corresponding positions after revising the manuscript. Please see lines 63. 

Reviewer 5 Report

I attach the opinion in pdf

Author Response

Response 1: Thank you for pointing this out. We have improved the clarity of Figure 1, and added the geographic coordinates and scale. Please see the revised manuscript, thank you again for your valuable suggestion.

Round 2

Reviewer 1 Report

I think that after the changes made, the article is more understandable.

Author Response

Response 1: Thank you for your constructive feedback. We are delighted to learn that the alterations we have made have enhanced the overall lucidity of the article. It was our privilege to include your insightful recommendations in our manuscript. Thank you.

Reviewer 2 Report

The authors ignored essential items required in the previous comments, which could inform us about the correctness of the presented results. 

1) Measurement time for acoustic and meteorological parameters is essential; it is difficult to trust a study with 10 seconds of measurement time.

2) Subjective data rating inversion indicated through several social sciences and statistical books was ignored. 

3) Observation of ANOVA assumptions was ignored.

4) In the latest version, the authors do not inform the procedures, software or algorithms used to calculate the UTCI.

Based on these facts, I recommend the rejection of this manuscript in the Forests journal.

The authors ignored essential items required in the previous comments, which could inform us about the correctness of the presented results. 

1) Measurement time for acoustic and meteorological parameters is essential; it is difficult to trust a study with 10 seconds of measurement time.

2) Subjective data rating inversion indicated through several social sciences and statistical books was ignored. 

3) Observation of ANOVA assumptions was ignored.

4) In the latest version, the authors do not inform the procedures, software or algorithms used to calculate the UTCI.

Based on these facts, I recommend the rejection of this manuscript in the Forests journal.

Author Response

Point 1: Measurement time for acoustic and meteorological parameters is essential; it is difficult to trust a study with 10 seconds of measurement time.

Response 1: Thank you for your valuable feedback on our manuscript. We acknowledge the importance of measurement time for both acoustic and meteorological parameters in our study. After careful verification, we found that the measurement time of 10 seconds is incorrect. We would like to express our sincerest apologies for the error. Our process for collecting experimental data involved asking subjects to walk along a designated route through various sensory zones. At each stop, the group would spend five minutes standing or sitting while carefully listening to sound elements and experiencing the temperature environment before completing a questionnaire online. Simultaneously, the main leaders were responsible for using equipment to collect data on temperature, global temperature, humidity, wind speed, and LAeq, also for a duration of five minutes[1]. In order to make the original text more objective and accurate, we have already revised the manuscript, please see Lines 270-278. Once again, we sincerely thank you for bringing this important aspect to our attention, and we are grateful for the opportunity to improve the quality and rigor of our manuscript.

  1. Gage, S.H.; Axel, A.C. Visualization of temporal change in soundscape power of a Michigan lake habitat over a 4-year period. Inform. 2014.

Point 2: Subjective data rating inversion indicated through several social sciences and statistical books was ignored.

Response 2: Thank you for your insightful comments regarding the omission of subjective data rating inversion in our study. We apologize for not addressing this aspect in our first revision. Upon careful consideration of your feedback, We have taken the advice and conducted a inversion of ratings and redrawn the graphs for acoustic comfort, acoustic preference, acoustic familiarity, overall acoustics, overall comfort, and overall satisfaction. This has been done to further analyze the data and provide a more comprehensive representation of the results. Please see Section 3.3 and 3.4. We appreciate your valuable input, which has significantly enriched our study.

Point 3: Observation of ANOVA assumptions was ignored.

Response 3:  We are sincerely grateful to the expert for raising this issue. We have made revisions in our latest manuscript and have provided a clear elucidation of the hypothesis verification prior to conducting variance analysis[1], please see Lines 290-297. Once again, we express our appreciation.

  1. Wang, F.; Huang, G.H.; Fan, Y.; Li, Y.P. Robust subsampling ANOVA methods for sensitivity analysis of water resource and environmental models. Water Resour. Manag. 2020, 34, 3199-3217.

Point 4: In the latest version, the authors do not inform the procedures, software or algorithms used to calculate the UTCI.

Response 4: We greatly appreciate the expert for raising this issue. We have incorporated the calculation process and software used for UTCI in the latest manuscript[1], please see Lines 201-211.  Thank you for your valuable suggestion.

  1. Peter, B.; Dusan, F.; Krzysztof, B.; Ingvar, H.; Gerd, J.; Bernhard, K.; Birger, T.; George, H. Deriving the operational procedure for the Universal Thermal Climate Index (UTCI). J. Biometeorol. 2012, 56.

Reviewer 4 Report

The revised version of the article is definitely better than the previous one. However, I suggest supplementing the paper with a few sentences of literature review on the soundscape of parks (this suggestion was included in the previous review). In addition, I suggest formulating the purpose of the paper more clearly. Unfortunately, it has not been changed from the previous version. Also, not all the proposed publications were cited by the authors. I have one more comment regarding the response to the previous review. I do not know why the Authors write the same thing twice in response to the general comment, point 3 (page 1-2) and comment 5 (page 3).

Author Response

Response : Thank you for this important suggestion.

  • As recommended, we have incorporated some references on soundscape in the latest manuscript and have included appropriate citations from expert-provided literature, please see Lines 60-69. We appreciate your invaluable suggestion.

  • Thank you for bringing up this question regarding the research purpose. We have further elaborated the purpose of the article in our latest manuscript with more details, please see Lines 131-142. Thank you for pointing it out.

  • Thank you for bringing up the issue of repetition. We will enhance our scrutiny and checks in the future to avoid any recurrence. We appreciate your pointing it out.